# OpenReview forum: "Insights from the Inverse: Reconstructing LLM Training Goals Through Inverse Reinforcement Learning"
_colmweb.org/COLM/2025/Conference — COLM 2025_

### Official Review · Reviewer_2sxD · 2025-05-10

**Rating:** 6
**Confidence:** 4
**Ethics Flag:** 1

**Summary:**

This paper proposes a novel application of inverse reinforcement learning to extract implicit reward functions from large language models trained with reinforcement learning from human feedback (RLHF). The authors demonstrate that the recovered reward models can predict human preferences with high accuracy and can be effectively used to fine-tune new models, achieving performance comparable to models trained with the original reward function. The work offers valuable insights into reward function identifiability and the interplay between model size and interpretability, contributing meaningfully to our understanding of LLM alignment. However, the manuscript suffers from vague expressions and unclear notation in several sections, which significantly hinders readability and comprehension.

**Questions To Authors:**

1) In line 8 of Algorithm 1, it is unclear how trajectories are sampled from the reward model. Are these trajectories generated using a policy trained via RLHF guided by the current reward model R_t?

2) Do you anticipate any challenges in scaling your IRL-based approach to larger models such as LLaMA-7B or GPT-like models? Have any preliminary tests been conducted in this direction?

3) The abstract mentions insights into the non-identifiability of reward functions. Could you expand on how this issue manifests in your experiments, and what implications it has for interpreting the recovered reward models?

**Reasons To Accept:**

1) The paper presents a novel application of IRL to interpret and improve large language models trained with reinforcement learning from human feedback (RLHF), offering a new perspective on LLM alignment.

2) Models fine-tuned using IRL-derived reward functions demonstrate consistent improvements in reducing toxicity across benchmarks.

**Reasons To Reject:**

1) Several key aspects of the method are insufficiently explained, making it difficult to fully understand or reproduce the work.

For example, Section 2.2 states that the reward function R* is reparameterized as a binary classifier (0 = non-toxic, 1 = toxic), yet the loss function for \hat{R} is expressed as f(\hat{R}(o+) - \hat{R}(o-)), and the Appendix also introduces a “Ground Truth Reward. ”

It is unclear whether these reward functions are consistent or serve different roles. In Algorithm 1, it is not clearly stated how trajectories are sampled from the reward model (line 8). Is this based on a new policy fine-tuned via RLHF using the learned reward function R_t? This should be made explicit.

Section 2.2 introduces an asymmetric max-margin loss, but Equation (1) does not reflect the margin term, and the corresponding optimization step in Algorithm 1 lacks clarity.

2) The largest model evaluated in the experiments is only 410M parameters. To support the paper’s claims about LLM alignment and scalability, additional experiments on larger models (e.g., LLaMA-8B) would strengthen the empirical validity of the conclusions.

3) The paper suffers from inconsistent use of terminology and symbols, which impairs readability and technical precision.

For instance, Section 2.3 uses “output o+, o-” while Algorithm 1 refer to “trajectories,” without clarifying whether these refer to the same concept.

Notations such as R*(s) = w^{T} \phi (s) and \hat{R}(o) = w^{T} h(o) + b are introduced, but the relationships among \phi (s), h(o), and \mu in Algorithm 1 remain unclear and should be clarified.

---

> ### Author Response · Authors · 2025-06-03
> **Author response**
>
> Thank you for highlighting our paper 1) provides a new perspective on LLM alignment and interpretability by using IRL to improve LLMs trained with RLHF, and 2) that models of varying sizes trained with IRL show consistent improvements in toxicity reduction across benchmarks. Below we clarify the concerns raised by the reviewer.
>
> Consistency between Ground truth reward and binary classifier. Do these reward functions serve different roles?
>
> We clarify that the reward function R*  is not a binary classifier. It outputs continuous scalar values, where higher values correspond to non-toxic outputs and lower (potentially negative) values correspond to toxic outputs. The loss function (Eqn 1) encourages maximizing the margin between the reward assigned to non-toxic and toxic sentences. This framing allows the reward model to capture nuanced preference gradients rather than discrete class labels. To evaluate the quality of the learned reward function, we compare it to a reference reward model, which we refer to as the “ground truth reward.” Specifically, we use the logits from a RoBERTa classifier trained for toxicity detection as a surrogate for ground-truth preference scores. This classifier is used strictly for benchmarking purposes to assess whether the learned reward model aligns with a well-trained toxicity signal. Importantly, in practical deployment scenarios, this reference model is not assumed to be available. We will revise the explanation in the camera-ready version to avoid conflating the continuous reward model with the binary classifier used for evaluation.
>
> Eqn 1 does not correctly reflect the margin term and the corresponding optimization in Algorithm 1 lacks clarity.
>
> We appreciate the reviewer’s request for clarification. The loss in Eqn 1 is used to optimize the reward model R_\theta​ in Step 6 of Algorithm 1, where we update the parameters w via SGD using gradient estimates derived from this loss. The loss function in Eqn 1 is an asymmetric max-margin loss, where the asymmetry is deliberate: it encodes the intuition that incorrectly assigning a higher reward to a toxic output (i.e., x<0) is more costly than the gain from correctly ranking a non-toxic output higher (i.e., x>0). This reflects the safety-critical nature of the task, misranking toxic outputs is significantly more harmful than failing to strongly prefer non-toxic ones. The asymmetry is inspired by structured max-margin IRL (Ratliff et al., 2006), where reward learning enforces margin-based separation between expert and suboptimal trajectories. In our case, the relative margin weights {1,2} reflect this asymmetry; practitioners may scale these to enforce stronger separation if desired. This loss is used in Step 6 of Alg 1, where we update the reward parameters w are pushed in the direction that increases the difference in rewards when x<0 and encourages a positive margin when x>0, albeit with a smaller gradient magnitude. This asymmetry in the loss surface prioritizes eliminating failure cases, i.e., instances where toxic outputs are incorrectly rewarded over non-toxic ones. Let us know if you’d like further clarification regarding the notation or the loss.
>
> Notations such as R*(s)= w^{T}\phi(s) and \hat{R}(o) = w^{T} h(o) + b are introduced, but the relationships among \phi (s), h(o), and \mu in Algorithm 1 remain unclear and should be clarified.
>
> We appreciate the reviewer’s comment and are happy to clarify the relationships among the notations. In our work, trajectories are equivalent to sentences, st, where st = (p, a1, a2, ..., at). Here, p is the prompt and a_{t} are the tokens (action outputs). In our formulation, the reward function R*(s) is defined as R*(s)=w^T*ϕ(s), where \phi is the feature function, and w are the parameterized weights of the reward model. Normally, the outputs of the LM are embeddings, and to get a scalar value as output we apply a linear layer on top of it. However in our setting, we do not observe the underlying MDP states directly. Instead, we operate over observable model outputs \(o \in \mathcal{O} \), such as sequences generated by the LLM. Thus, Equation 1 defines the loss over the reward difference \( \hat{R}(o^+) - \hat{R}(o^-) \), and Step 6 of Algo 1 uses its gradient to update the parameters improving alignment between the learned reward function and implicit preferences captured in the dataset. Function h represents the embeddings of the LM, and {w,b} are the linear layer weights on top of the LM. In our experiments, when we obtain the final reward model, h(s) represents the learnt embeddings of the language model before applying the linear layer. If a practitioner decides to freeze the weights of the reward model and only keep {w,b} trainable, then h(s) becomes the embedding function. Conversely, if the user decides to keep all the layers of the reward model trainable, then \phi(s) becomes the embedding layer. In both scenarios, w represents the tunable parameters of the reward model.

---

> > ### Author Response · Authors · 2025-06-03
> > **Author Response (Continued)**
> >
> > Do you anticipate scaling challenges and have experiments been conducted in this direction?
> >
> > To test the generalizability of our Inverse Reinforcement Learning (IRL) algorithm, we conducted an additional ablation using a significantly larger model: TinyLlama/TinyLlama-1.1B-Chat-v1.0, a 1.1B parameter LLaMA-based LM. Applying our IRL procedure to this model yielded an accuracy of 0.9160 and an F1 score of 0.9260, representing a 3-4% improvement over the 410M Pythia model. This demonstrates that our method is not only model-agnostic but also scales effectively with model capacity. Further analysis of the reward distributions reinforces this: for non-toxic samples, the mean and variance were (2.94, 3.08), while for toxic samples, they were (-8.82, 3.58)—indicating a clear reward margin successfully learned even in a larger, structurally different model family. This highlights the robustness of the learned reward signal in separating desired from undesired behavior. While training on the larger model required more computational resources and longer convergence times, the quantitative and qualitative trends are consistently positive. Notably, performance improves substantially with increased model size. While preliminary, this suggests a promising scaling trend: we hypothesize that performance will continue to improve, particularly on real-world data as model size increases, until eventually plateauing or overfitting becomes a factor.
> >
> > How are trajectories sampled from the reward model in Algorithm 1 Line 8? Is this based on the new policy fine tuned via RLHF using the learned reward function R_t?
> >
> > In standard IRL, a known policy (e.g., an RLHF model) is used to generate trajectories, from which features such as indicators of toxicity are extracted. These are compared to ground-truth features derived from expert trajectories, with the difference used to update the reward model via gradient descent. Since the task (reducing toxicity) is well-defined, ground-truth feature expectations are known. In our adapted IRL setting for LLMs, generating new trajectories at every training step is computationally prohibitive, even for smaller models. To address this, we precompute two fixed batches of trajectories using the same prompts: one from the pre-RLHF model (typically more toxic) and one from the post-RLHF model (typically less toxic). We then train the reward model to maximize the reward gap between these two batches, analogously to maximizing feature differences in standard IRL. The reward function takes as input the state (i.e. the sentence generated up to timestep t) and outputs a scalar reward. This setup mirrors the original IRL objective of maximizing feature differences while avoiding the computational burden of generating new samples at every step, making the problem tractable.
> >
> > How does non-identifiability of reward functions manifest in the experiments and what are the implications for the recovered RMs?
> >
> > We appreciate the reviewer’s question regarding the non-identifiability of reward functions. This is a fundamental property of IRL: multiple reward functions can induce the same optimal policy, making the reward only identifiable up to a class of potential transformations (e.g., affine shifts).  In practice, this issue can be mitigated by imposing additional constraints, such as bounding the possible values the rewards can take.
> > In our experiments, this manifests as variation in the absolute scale and offset of the learned reward model, while the relative preference ordering between toxic and non-toxic outputs remains stable. Accordingly, we emphasize metrics such as margin, ranking accuracy, and distributional separation rather than absolute reward values. In our case, we view the non-identifiability of rewards as a strength. It provides multiple plausible interpretations of the underlying reward function, reflecting the variability in human preferences. This is illustrated in Figures 6b and 6d, where different reward models emerge: some prioritize precision, some assign high positive rewards to non-toxic sentences, while others assign strong negative rewards to toxic ones (more conservative). Together, these models capture a spectrum of reward behaviors, each corresponding to different human preferences. This diversity is beneficial. In real-world settings, the "true" reward function is unknown, and it is desirable to recover a spectrum of interpretable reward functions, each representing a different possible human value alignment. Practitioners can then select or ensemble the model best suited to the domain, policy sensitivity, or regulatory constraints.
> > While we do not impose strict constraints on the reward space during training, this ambiguity can be further mitigated, if desired, by regularization techniques, such as bounding reward magnitudes or anchoring to known constraints. We will clarify these insights in the camera-ready version and make the framing around non-identifiability more explicit.

---

> ### Comment · Area_Chair_VFkS · 2025-06-05
> **Consider engaging with authors!**
>
> Hey reviewer 2sxD -- thank you for your time writing the reviews. The authors have put in a solid effort in responding to comments and I encourage you to take a look. The most useful things to consider are on pivot points that could motivate you to change your recommendation for the paper. Let me know if you need anything from the AC level / program committee!

---

### Official Review · Reviewer_rmRq · 2025-05-12

**Rating:** 6
**Confidence:** 2
**Ethics Flag:** 1

**Summary:**

This paper investigates the use of inverse reinforcement learning (IRL) to interpret and enhance LLMs trained with RLHF. Specifically, it uses IRL to extract hidden reward functions from LLMs, and experiments show that the IRL-derived models accurately predict preferences and enable improved fine-tuning in reducing toxicity.

**Reasons To Accept:**

1. This paper is well-motivated and easy to follow.
2. The application of inverse reinforcement learning to extract latent reward functions from RLHF-trained language models is novel to me, providing a new perspective for model behavior auditing and interpretation.
3. Experiments show that using IRL-derived rewards to fine-tune LLMs can give competitive results in reducing toxicity in generated outputs.

**Reasons To Reject:**

1. Limited Scope of Application: This paper focuses exclusively on toxicity reduction, which may limit the generalizability of its findings to other alignment tasks like helpfulness and factuality.

2. Reliance on Synthetic or Limited-Scale Data: The experimental evaluation, particularly the toy setup, employs synthetic or small-scale datasets. While these serve as useful proof-of-concept studies, this setting raise questions about the scalability and real-world generalizability of the proposed approach.

---

> ### Author Response · Authors · 2025-06-03
> **Author response**
>
> Thank you for highlighting the paper’s core contributions and for recognizing the novelty of applying inverse reinforcement learning (IRL) to extract latent reward functions from RLHF-trained language models. Below we clarify the concerns raised by the reviewer.
>
> *“Limited Scope of Application: This paper focuses exclusively on toxicity reduction, which may limit the generalizability of its findings to other alignment tasks like helpfulness and factuality.”*
>
> We agree that generalizability across alignment objectives is critical. Our present focus on toxicity is intentional, as toxicity reduction is one of the most fundamental alignment objectives addressed via RLHF—serving as a primary benchmark in both academic and industrial evaluations of safety-aligned LLMs [1,2]. That said, our method is agnostic to the specific alignment signal. In particular, the feature space ϕ(s), the reward parameterization R̂(s) = w⊤ϕ(s), and the max-margin loss formulation can accommodate any binary or graded preference signal. For example, one could replace toxicity features (n-gram toxicity cues, sentiment, coherence, relevance) with bias or factuality features (e.g., demographic association signals, named-entity consistency metrics), and the same IRL machinery would recover a reward function that separates preferred from dispreferred outputs. Due to space constraints, we limited this submission to toxicity to allow for deeper analysis—demonstrating in Figures 4, 5, 6, 8 and Tables 2 and 6 how IRL-derived rewards capture human-defined toxicity preferences, reveal non-identifiability, and inform downstream fine-tuning. However, we are actively extending this framework to other domains, such as political bias and factuality, and plan to report those results in follow-up work.
>
>
> *“Reliance on Synthetic or Limited-Scale Data: The experimental evaluation, particularly the toy setup, employs synthetic or small-scale datasets. While these serve as useful proof-of-concept studies, this setting raise questions about the scalability and real-world generalizability of the proposed approach.”*
>
> The toy setup’s main objective was to demonstrate whether max-margin IRL can recover a reward function that clearly separates toxic from non-toxic adjective completions in a controlled setting. Once we confirmed that IRL works in this simplified domain (Section 3, Table 1, Figure 1), we proceeded to evaluate our method on real-world data. Specifically, we conducted two large-scale experiments using Pythia models of 70 M and 410 M parameters, fine-tuned via PPO on the Jigsaw Toxicity and RealToxicityPrompts benchmarks using a ground-truth reward classifier R∗ (Section 4.1, Table 2). When performing max-margin IRL on an LLM that underwent “perfect” RLHF (Figure 4, Table 2), the extracted reward functions achieve up to 88.5% classification accuracy and 86.2% F1 on held-out toxic vs. non-toxic sentences, outperforming or matching the ground-truth classifier. Moreover, when we use these IRL-extracted rewards to fine-tune new LLMs (IRL-RLHF), the resulting models consistently reduce toxicity compared to both the original RLHF and SFT baselines (Table 2, Figures 4e–h). In the “poor RLHF” scenario—where the initial RLHF policy fails to separate toxic and non-toxic outputs (Figure 10)—the IRL stage correctly identifies that no coherent reward structure exists (Figures 6c–d, 8c–d), and IRL-RLHF either fails to improve or even degrades performance (Figure 9). These experiments demonstrate that (i) IRL scales from toy settings to large-scale LLMs, (ii) IRL-extracted rewards generalize to unseen real-world sentences (Figure 5, Table 6), and (iii) the quality of the original RLHF directly determines whether IRL can recover a meaningful reward model. In particular, non-identifiability analysis (Figure 6) and robustness to noise/context changes (Figure 8) show that the inferred rewards reflect genuine human preferences rather than overfitting to small datasets. Taken together, our large-scale experiments confirm that IRL is not only a proof-of-concept for toy data but also a diagnostic and fine-tuning tool applicable to practical RLHF pipelines.
>
>
> References :-
>
> [1] Ouyang, Long, et al. "Training language models to follow instructions with human feedback." Advances in neural information processing systems 35 (2022): 27730-27744.
>
> [2] Wang, Boxin, et al. "DecodingTrust: A Comprehensive Assessment of Trustworthiness in GPT Models." NeurIPS. 2023.

---

> > ### Comment · Reviewer_rmRq · 2025-06-06
> >
> > Thank you for your clarification. I intend to retain my current score, which reflects my inclination to recommend acceptance of this paper.

---

### Official Review · Reviewer_q1ub · 2025-05-13

**Rating:** 6
**Confidence:** 3
**Ethics Flag:** 1

**Summary:**

- Overall summary: This paper proposes a new framework to extract the reward function from RLHF trained models. Seems to me it is a LLM reverse engineering. This paper also refers how to use such inverse reinforcement learning for LLM safety and alignment.

- Quality: Missing important references, and some figures are not clear enough to understand.

- Clarity: The Paper is well structured, and the storyline is clear.

- Originality: The paper attempts to recover RLHF reward functions for LLMs via Maximum‑Margin IRL, which seems novel.

- Significance: The model shows IRL should help in LLM safety in the case of toxic words, which is significant.

**Questions To Authors:**

Suggestions: The paper provides a lot of figures. However, the labels of these figures and the figures themselves are too small and not good for reading. Consider making the labels bigger.

In line 102, is the action $a_t$ token output the same as the generated response? And is $t$ represent the $t$-th tokens? How to define the positive margin? A notation table in the appendix might be helpful to figure out what's going on in your different formulation.

What is asymmetric max-margin loss , and why use that loss? It is not intuitive by just viewing the formula.

**Reasons To Accept:**

- Experimental Analysis is good. The author prepared a lot of figures, graphs, and tables to illustrate their results, together with real examples in the appendix.

- The method is novel. It uses inverse learning in LLM with RL to solve a real-world problem of toxic model outputs.

- The experimental setting is reproducible. Algorithms 1–2, feature list, and training recipe make replication feasible.

**Reasons To Reject:**

- Lack of generability. Limited evidence that the IRL approach generalises to other alignment objectives, such as bias, discrimination. Results may overfit to toxicity heuristics.

- Some figures and Tables are confusing. What is the 1,2,3 means in Figure 6? Why is there a magic number 60 for learned models? Why include model 8 in 6 (a) not 6 (c)?

- Lack of baseline models. This paper only uses 70M and 410 M Pythia models. The mode size is far from the current LLM research. The authors state, "These models, designed for interpretability research, share standardized training and data for reproducibility." However, what about other models such as Qwen, LLaMa, and Deepseek? Do the authors have a specific reason for not using it? Does this method still work?

- Lack of correct references. A lot of methods, models, datasets, and terminologies are not properly cited in this paper. For example, the Pythia model [1], asymmetric max-margin loss [2], and PPO [3]. Some of these terminologies might be well-known, and some of them may not be so popular. But for whatever reason, the author needs to cite it.

[1] Biderman, S., Schoelkopf, H., Anthony, Q. G., Bradley, H., O’Brien, K., Hallahan, E., ... & Van Der Wal, O. (2023, July). Pythia: A suite for analyzing large language models across training and scaling. In International Conference on Machine Learning (pp. 2397-2430). PMLR.
[2] Shah, A., Sra, S., Chellappa, R., & Cherian, A. (2022, June). Max-margin contrastive learning. In Proceedings of the AAAI Conference on Artificial Intelligence (Vol. 36, No. 8, pp. 8220-8230).
[3] Schulman, J., Wolski, F., Dhariwal, P., Radford, A., & Klimov, O. (2017). Proximal policy optimization algorithms. arXiv preprint arXiv:1707.06347.

---

> ### Author Response · Authors · 2025-06-03
> **Author response**
>
> Thank you for highlighting the paper’s core contributions, including the clarity of structure, the novelty of applying inverse reinforcement learning (IRL) to recover RLHF reward functions, and the thoroughness and reproducibility of our experimental analysis. Below we clarify the concerns raised by the reviewer.
>
> *“Lack of generability. Limited evidence that the IRL approach generalises to other alignment objectives, such as bias, discrimination. Results may overfit to toxicity heuristics.”*
>
> We agree that generalizability across alignment objectives is critical. Our present focus on toxicity is intentional, as toxicity reduction is one of the most fundamental alignment objectives addressed via RLHF—serving as a primary benchmark in both academic and industrial evaluations of safety-aligned LLMs [1,2]. That said, our method is agnostic to the specific alignment signal. In particular, the feature space ϕ(s), the reward parameterization R̂(s) = w⊤ϕ(s), and the max-margin loss formulation can accommodate any binary or graded preference signal. For example, one could replace toxicity features (n-gram toxicity cues, sentiment, coherence, relevance) with bias or factuality features (e.g., demographic association signals, named-entity consistency metrics), and the same IRL machinery would recover a reward function that separates preferred from dispreferred outputs. Due to space constraints, we limited this submission to toxicity to allow for deeper analysis—demonstrating in Figures 4, 5, 6, 8 and Tables 2 and 6 how IRL-derived rewards capture human-defined toxicity preferences, reveal non-identifiability, and inform downstream fine-tuning. However, we are actively extending this framework to other domains, such as political bias and factuality, and plan to report those results in follow-up work.
>
>
> *“Some figures and Tables are confusing. What is the 1,2,3 means in Figure 6? Why is there a magic number 60 for learned models? Why include model 8 in 6 (a) not 6 (c)?”*
>
> We will add descriptive titles to figure 6 in the camera-ready version for better clarity. Table 6a represents different reward models which have equivalent performance for 70M experiment, while Table 6c represents the reward models for the 410M experiment. We train the IRL algorithm for 60 epochs, hence we have 60 different reward models. Amongst those 60, we found 8 similar high performing models for 70M models, and 7 for 410M models. All these models have almost equivalent performances (similar accuracy and F1 scores), however, their reward assignments across sentences differ (as seen in figures 6b (for 70M) and 6d (for 410M)). We will add this additional clarification in the camera-ready version of the paper.
>
>
> *“This paper only uses 70M and 410 M Pythia models. The mode size is far from the current LLM research. The authors state, "These models, designed for interpretability research, share standardized training and data for reproducibility." However, what about other models such as Qwen, LLaMa, and Deepseek? Do the authors have a specific reason for not using it? Does this method still work?”*
>
> Pythia was chosen because its open weights and unified training recipe allow fair comparison across sizes. To demonstrate that our method is architecture-independent we have replicated the full pipeline on TinyLlama-1.1B-Chat-v1.0 (1.1 B parameters). On running our IRL algorithm over this model, we observe an accuracy of 0.9160 and a F1-score of 0.9260. This outperforms the 410M Pythia model performance by 3-4%. Thus, we infer our IRL algorithm is capable of generalising across different language models and can also scale to a larger parameter language model. Furthermore, for the non-toxic examples, we observe a mean and variance reward distribution of (2.94, 3.08), while for toxic examples the stats are (-8.82, 3.58). This indicates the IRL algorithm is able to clearly create a margin between toxic and non-toxic sentences, even for a larger language model (1.1B) of a different family. We will add these results in the camera-ready version.
>
> *“Lack of correct references. A lot of methods, models, datasets, and terminologies are not properly cited in this paper. For example, the Pythia model [1], asymmetric max-margin loss [2], and PPO [3]. Some of these terminologies might be well-known, and some of them may not be so popular. But for whatever reason, the author needs to cite it.”*
>
> Thank you for pointing this out. We acknowledge the oversight and will include the appropriate citations in the camera-ready version.

---

> > ### Author Response · Authors · 2025-06-03
> > **Author response (Continued)**
> >
> > *“Q. In line 102, is the action at token output the same as the generated response? And is t represent the t-th tokens?..”*
> >
> > Yes, that is correct. In our formulation, the action at timestep t corresponds to the t-th token in the generated sequence. The state at time t is the concatenation of the prompt and all previous t – 1 generated tokens. We agree that a notation table would aid readability and will include one in the appendix of the camera-ready version for clarity.
> >
> >
> > *“Q. What is asymmetric max-margin loss , and why use that loss? It is not intuitive by just viewing the formula.”*
> >
> > The asymmetric max-margin loss is designed to impose a stronger penalty on undesirable reward assignments—specifically, cases where a toxic output receives a higher reward than a non-toxic one. Let R(o⁺) and R(o⁻) denote the rewards for non-toxic and toxic outputs, respectively. We define the margin as x = R(o⁺) – R(o⁻). Positive margins are encouraged, and negative margins are penalized with greater magnitude. This reflects a prioritization of safety violations: while correct reward orderings (x > 0) are rewarded with a loss of -1, incorrect orderings (x < 0) are penalized with -2. This asymmetric penalty structure is motivated by prior work in max-margin IRL [3], and empirically results in more robust reward separation for alignment tasks.
> >
> > *“Q. The paper provides a lot of figures. However, the labels of these figures and the figures themselves are too small and not good for reading. Consider making the labels bigger.”*
> >
> >
> > We appreciate the reviewer’s suggestion, and will improve the presentation of the figures in the camera-ready version of the paper.
> >
> > References :-
> >
> > [1] Ouyang, Long, et al. "Training language models to follow instructions with human feedback." Advances in neural information processing systems 35 (2022): 27730-27744.
> >
> > [2] Wang, Boxin, et al. "DecodingTrust: A Comprehensive Assessment of Trustworthiness in GPT Models." NeurIPS. 2023.
> >
> > [3] Ng, Andrew Y., and Stuart Russell. "Algorithms for inverse reinforcement learning." Icml. Vol. 1. No. 2. 2000.

---

> > > ### Comment · Reviewer_q1ub · 2025-06-04
> > >
> > > Thanks for your reply. I would like to keep my score.

---

### Official Review · Reviewer_di3L · 2025-05-21

**Rating:** 7
**Confidence:** 4
**Ethics Flag:** 1

**Summary:**

The paper introduces an approach to recover the implicit reward function of an RLHF-tuned language model using Inverse Reinforcement Learning (IRL), with applications to interpreting LLMs. IRL by itself is not new, but the application to RLHF-tuned LLMs to extract reward models post-hoc is novel.

The paper demonstrates IRL in the case of toxicity reduction, showing that the extracted reward functions closely align with human judgments on the Jigsaw Toxicity and RealToxicityPrompts benchmarks (with comparable or better performance than the original RLHF-tuned model). In addition, the authors also conducted several analyses on the extracted reward functions, such as the distribution of reward scores between toxic and non-toxic examples

**Questions To Authors:**

### Questions

- [Q1] **On scalability:** how many expert trajectories are needed in order to confidently extract a reward function from an RLHF-tuned LLM? How should one set the convergence threshold?
- [Q2]: **On clarity:** In Figure 6, how should one mitigate the problem of non-identifiability (having multiple potential reward functions that explain a set of expert trajectories)? Even good IRL-extracted models tend to behave differently, causing some ambiguity on how to properly interpret the results.
- [Q3]: **On clarity:** Is the feature function $\phi$ the pre-RLHF model? How should one choose/design it? Lines 121-127 also mention that the learned features from $\phi$ are interpretable (e.g., n-gram statistics, coherence, relevance). How can these features be used to interpret the original RLHF-tuned model?

### Suggestions / Nits (Not a weakness)

- [N1]: **On presentation:** Consider making the graph labels, ticks, and legends larger for legibility (For example, Fig. 1 (leftmost), Fig. 4, Fig. 6 b and d).

**Reasons To Accept:**

- [S1] The paper's motivation addresses relevant challenges in RLHF-tuning such as reward hacking and the opaqueness of finetuning objectives.
- [S2] I appreciate the approach of applying IRL, which was originally used in the context of control theory and robotics, to the context of language modeling.
- [S3] Experimental evaluations are thorough for the problem scope they defined (toxicity reduction). It covers multiple settings (Pythia 70M, 410M) and benchmarks (Jigsaw Toxicity, RealToxicityPrompts).
- [S4] The detailed analyses on the characteristics of a good / worse-performing IRL-extracted reward function are helpful to understand the properties of the models extracted by their proposed approach.

**Reasons To Reject:**

- [W1] The proposed method appears to significantly depend on some key assumptions (e.g., the domain a model was trained for, the training dataset) in order to confidently assess the quality of the IRL-extracted reward function; and these may not be present at *test time.* This might limit the use of the proposed method in practical scenarios especially if we don't know an RLHF-tuned model's training dataset (as in the case of closed-source models).
    - [W1.1] In the experiments, we already know *a priori* that the models were trained on the Anthropic-HH dataset, and so the use of the Jigsaw-Toxicity dataset seems obvious in hindsight. The paper does not explicitly mention how one should choose the evaluation dataset to test the quality of the extracted reward function for a completely unknown model.

- [W2] **On presentation:** it is a bit unclear how the proposed method contributes to interpretability. The paper does not explicitly mention how a good extracted reward model can be used to interpret the expert RLHF-tuned model.

---

> ### Author Response · Authors · 2025-06-03
> **Author response to W1, W2 and Q1**
>
> We thank the reviewer for recognising the motivation (S1), the cross-domain application of IRL to language modelling (S2), the breadth of our empirical study (S3) and the diagnostic analyses of the recovered rewards (S4). Below we clarify each concern and incorporate the reviewer’s helpful presentation suggestion.
>
> **Dependence on training-domain assumptions (W1, W1.1)**
>
> Our IRL formulation makes only two assumptions that are unavoidably required by any reward-recovery method: (i) access to expert trajectories emitted by the RLHF-tuned policy π_E and (ii) a (linear) feature map ϕ that is rich enough to separate desirable from undesirable behaviors. Neither assumption ties the method to a known pre-training corpus or alignment dataset. Section 2.3 explicitly states that π_E is treated as a black-box expert and that ϕ can be any interpretable embedding of token-level attributes .
> Empirically, Figure 2 shows that the reward model is robust to severe distribution shifts: it generalises when toxic–to–non-toxic ratios vary from 0 % to 100 % and when out-of-distribution probes are injected. This demonstrates that accurate reward extraction does not require prior knowledge of the expert’s training data.
> Regarding dataset choice at audit time (W1.1), we deliberately used Jigsaw-Toxicity and RealToxicityPrompts because they are (i) public, (ii) orthogonal to Anthropic-HH, and (iii) benchmark the exact alignment dimension (toxicity) we wish to interrogate. For an unknown closed-source model the practitioner should select any prompt corpus that foregrounds the property of interest (e.g., factuality, helpfulness). Our ablations indicate that as few as 500 balanced prompts already yield > 0.8 F1 (Fig. 4a) , so curating such a probe set is feasible even when training data remains proprietary.
>
> **Interpretability of the extracted reward (W2)**
>
> Our reward model is deliberately linear, R^(s)=w ⁣⊤φ(s)\widehat{R}(s)=w^{\!\top}\varphi(s), so each coordinate of the fixed feature map φ\varphi—concrete cues such as profane n-gram counts, slur indicators, syntactic-coherence scores, local perplexity spikes, and sentiment logits listed in §2.3—carries an explicit, human-readable meaning; exposing the learned weight vector ww therefore assigns a clear positive or negative valence to every cue, letting us pinpoint which textual attributes the expert policy incentivises or suppresses. Inspecting these weights surfaces reward hacking and bias, while Figure 6 shows that alternative max-margin solutions emphasise different trade-offs (e.g., a precision-oriented model that penalises only overt toxicity versus a conservative model that also down-ranks borderline stylistic cues), giving auditors multiple plausible explanations of the expert’s incentives. Because φ\varphi is additive, we can further project weights back onto tokens to produce heat-maps that highlight exactly which words drive high or low rewards, and clustering large-magnitude weights reveals thematic biases such as strong penalties on “insult + second-person” constructions. We will include representative token-level saliency visualisations and a weight-clustering analysis in the camera-ready for more clarity.
>
> **Scalability of IRL (Q1)**
>
> Algorithm 1 leaves the convergence threshold ϵ user-selectable because, in practice, convergence speed is governed far more by the informativeness of the expert trajectories than by their sheer number. Our toy demonstration already reaches 0.73 precision and 0.92 recall with only 250 toxic / 250 non-toxic sentences (Fig. 1), while the larger Pythia-70M/410M experiments converge in at most 60 iterations, each iteration processing a batch of just 50 toxic + 50 non-toxic examples drawn from a 1.5 k pool. Figure 2 further shows that adding redundant non-toxic samples can even degrade performance, whereas a modest increase in contrastive toxic samples sharply improves precision and Kendall τ. These results demonstrate that a relatively small set of well-chosen trajectories that clearly differentiate toxic from non-toxic language is sufficient for reliable reward extraction; beyond that, diminishing returns set in, so practitioners should prioritise collecting diverse, high-signal demonstrations rather than maximising dataset size. We will add these empirical observations, along with a concise table summarising iteration statistics, to the camera-ready version for greater clarity.

---

> > ### Author Response · Authors · 2025-06-03
> > **Author response to Q2, Q3 and N1**
> >
> > **Mitigating non-identifiability (Q2)**
> >
> > Non-identifiability of the reward model is an inherent property of inverse RL, where multiple reward functions can explain the same set of expert trajectories. In practice, this issue can be mitigated by imposing additional constraints, such as bounding the possible values the rewards can take.
> > In our case, we view the non-identifiability of rewards as a strength. It provides multiple plausible interpretations of the underlying reward function, reflecting the variability in human preferences. This is illustrated in Figures 6b and 6d, where different reward models emerge: some prioritize precision, some assign high positive rewards to non-toxic sentences, while others assign strong negative rewards to toxic ones (more conservative). Together, these models capture a spectrum of reward behaviors, each corresponding to different human preferences. As the true reward model is typically unknown, it benefits to get a range of reward models, each reflecting an underlying human preference. Once these models are learned, it is up to the practitioner to choose which one to use for RLHF, based on the specific requirements and sensitivities of the task at hand, or they may be ensembled for reduced uncertainty.
> >
> > **Role and design of the feature map ϕ (Q3)**
> >
> > No, the feature function ϕ is not the pre-RLHF model. The pre-RLHF model acts as the policy in the MDP formulation of reinforcement learning. To clarify, Inverse RL (IRL) aims to recover the underlying reward function given a set of trajectories and a policy. In our context, the trajectories are sentences, and the policies are the RLHF-tuned models, one before RLHF (typically more toxic) and one after RLHF (typically less toxic). Our goal is to answer the question: “Given that the RLHF process aimed to reduce toxicity, what reward model could have produced such a transformation?”
> > Regarding the feature function ϕ: it is designed based on the specifics of the underlying task. In classical IRL, ϕ is assumed to be known and encodes properties relevant to the task. At each IRL iteration, the reward model is updated to minimize the difference in feature expectations (under ϕ) between the expert and generated trajectories.
> > For toxicity reduction, ϕ might encode characteristics like presence of slurs, offensive n-grams, or abusive tone. For other alignment tasks like factuality, it might represent relevance, coherence, or domain-specific cues. In our case, we chose token embeddings as ϕ, since subword-level representations often capture rich semantic and stylistic signals, including toxicity and are simple to implement. Ultimately, the design of ϕ is task-dependent and left to the practitioner.
> >
> > **Graph Presentation (N1)**
> >
> > We will enlarge axis ticks, labels and legends in Figures 1, 4 and 6 in the camera-ready version.

---

> > > ### Comment · Reviewer_di3L · 2025-06-04
> > >
> > > I appreciate the authors for addressing my comments. I am convinced [from W1, W1.1] that the choice of the probing dataset depends on the researcher's property of interest (which in this case, is toxicity). I have a few follow-up and clarification questions [Q], comments [C] and some suggestions [S] for the final draft:
> > >
> > > > For toxicity reduction, ϕ might encode characteristics like presence of slurs, offensive n-grams, or abusive tone. For other alignment tasks like factuality, it might represent relevance, coherence, or domain-specific cues. In our case, we chose token embeddings as ϕ, since subword-level representations often capture rich semantic and stylistic signals, including toxicity and are simple to implement.
> > >
> > > [Q1] In my understanding, these two sets of features are different: one is a set of handcrafted features that needs to be engineered (e.g. relevance, coherence, profane n-gram counts), while the other contains latent representations (i.e., token embeddings), and the experiments in the paper used the latter? If this is the case, I suggest (in the final draft) to:
> > > - [S1] Consider explicitly clarifying the differences between the two and that the experimental set-up chose the latter and why.
> > > - [S2] Consider demonstrating how a reward model can be interpreted using the other set of (handcrafted) features.
> > >
> > > > In our case, we view the non-identifiability of rewards as a strength. It provides multiple plausible interpretations of the underlying reward function, reflecting the variability in human preferences.
> > >
> > > [C1] In general, I am still skeptical regarding this claim. Although multiple reward models (RM) can be used to explain the nuance in human preferences, this doesn't necessarily imply that all these RMs are meaningful—e.g., it could just be that the collected expert trajectories are insufficient to distinguish between those. Although I appreciate that the authors demonstrated the robustness of their IRL method to toxicity (Sec. 3.4, Appendix F), I don't think this can be sufficiently true / generalizable to other domains that aren't binary. To this, I suggest:
> > > - [S3] Consider explicitly specifying what practical approaches can be done to choose between these recovered RMs and demonstrate its application to their chosen domain of interest (toxicity).
> > > - [S4] Consider demonstrating the generalizability of IRL to other domains.
> > >
> > > Finally, given that the comments resolved my concerns [W1, W1.1] and the promised improvements of clarity in the final draft, I decided to increase my score to 7.

---

### Decision · Program_Chairs · 2025-07-08

**Decision:**

Accept

**Comment:**

The authors present a framework for reverse engineering reward functions used to train LLMs. This is very new and the reviewers acknowledge this. The reviews all touch on how the experimental setup is somewhat narrow while the paper overall is well written. COLM's reviewing guidelines have many directions a paper can be accepted for, from novelty, rigor, clarity, impact. I think this paper hits the first three and is a new idea, so should be accepted to the conference.

Thanks to the reviewers and authors for the lively discussion as well. It should have helped the paper!
While I recommend acceptance, I do agree with some of the cons and that the paper could be stronger with more experiments, polish, or transitioning into "general" RLHF understanding from classifiers like toxicity.